# Preparation of Ti-Al-Si Gradient Coating Based on Silicon Concentration Gradient and Added-Ce

**Zihan Wang, Faguo Li \*** , **Xiaoyuan Hu, Wei He, Zhan Liu and Yao Tan**

School of Materials Science and Engineering, Xiangtan University, Xiangtan 411105, China;
wangzihann0524@163.com (Z.W.); hxyajr@outlook.com (X.H.); 15750022348@163.com (W.H.);
liuzhanxtu@yeah.net (Z.L.); aptx486912010504@163.com (Y.T.)
**\*** Correspondence: lifaguo@xtu.edu.cn

**Abstract:** Titanium and titanium alloys have excellent physical properties and process properties and are widely used in the aviation industry, but their high-temperature oxidation resistance is poor, and there is a thermal barrier temperature of 600 °C, which limits their application as high-temperature components. The Self-generated Gradient Hot-dipping Infiltration (SGHDI) method is used to prepare the Ti-Al-Si gradient coating based on the silicon concentration with a compact Ti(Al,Si)$_3$ phase layer, which can effectively improve the high-temperature oxidation resistance of the titanium alloy. Adding cerium can effectively inhibit the generation of the $\tau_2$: Ti(Al$_x$Si$_{1-x}$)$_2$ phase within a certain hot infiltration time so as to form a continuous dense Al$_2$O$_3$ layer to further improve the oxidation resistance of the coating. Studies have found that multiple Ti-Al binary alloy phase layers are formed during the high-temperature oxidation process, which has the effect of isolating oxygen and crack growth, and effectively improving the high-temperature resistance of the coating oxidation performance.

**Keywords:** titanium alloy; Ti-Al-Si gradient coating; cerium; self-generated gradient hot-dipping infiltration (SGHDI) method; microstructure and properties



## 1. Introduction

High-temperature titanium alloys are important structural materials for the key components of modern aero-engines, which are known as the 'pearl in the crown' of industrial manufacturing. Aero-engine is a concentrated embodiment of a country's comprehensive national strength, industrial foundation and technological level, and it is an important guarantee of national security. As the aviation industry develops rapidly, the thrust-to-weight ratio of an areo-engine is required to be higher. Titanium alloys can be used in fan and compressor parts of aero-engines instead of Ni-based superalloys, such as compressor discs, vanes, navigations and adapter rings, which can reduce the weight of the compressor by 30–35% [1]. At present, the amount of titanium alloys has accounted for 25–40% of the total weight of advanced aero-engines. However, traditional titanium alloys have to be used in air for a long time at the temperature of 540–600 °C or so, above which will result in excessive oxidation and oxygen brittleness caused by oxygen diffusion to the substrate [2]. At present, the maximum short-term operating temperature of mature high-temperature titanium alloys is 600–750 °C. As the temperature rises, the high-temperature creep and high-temperature oxidation resistance of titanium alloys will significantly decrease, which has become a bottleneck restricting the development of titanium alloys' application at higher temperatures [3]. Thermal corrosion is a phenomenon in which oxygen and other corrosive gases work together with salts deposited on the surface of materials in a high-temperature environment to accelerate corrosion, and its harmfulness is much greater than thermal oxidation. In addition, the research on high-temperature thermal corrosion resistance of high-temperature coatings in service above 1000 °C will be the focus of research in the future [4]. Therefore, in order to further improve the thrust–weight ratio of

aero-engines and meet the requirements of materials for high-performance aero-engines, it is imperative to develop a new high-temperature titanium alloy coating with higher operating temperature.

For decades, people have studied coating elements such as Al, Si, Cr, Nb, C, S and Mo, and found that aluminum was the most effective among many anti-oxidation elements [5]. Aluminum reacts at high temperatures to produce stable $Al_2O_3$ [6], which has an excellent protective effect on the metal substrate. Therefore, aluminum is the preferred element for high-temperature oxidation resistance coatings [7]. Among the Ti–Al system's intermetallic compounds, only the $TiAl_3$ can form dense $Al_2O_3$ film in the air, whose Al content is 75% and oxidation resistance is good [8]. High-temperature diffusion aluminizing of titanium alloys can form a $TiAl_3$-rich coating, which can greatly improve its oxidation resistance [9]. The commonly used diffusion method is the hot-dipping method, which was initially applied to steel materials [10] and is often used to improve the corrosion resistance, abrasive resistance and oxidation resistance at high temperature. Studies have found that the hot-dipping method can also improve the performance of titanium alloys [11–13]. However, aluminum hot-dipping cannot provide effective protection for titanium alloy above 800 °C. In addition, because multiple Ti-Al phase layers are generated, there will be penetrating cracks because the thermal expansion coefficient of the phase layer does not match the substrate [11]. However, adding silicon can reduce the number of transverse cracks in the coating. Compared with the hot-dip aluminizing layer, the Ti-al-si layer is tough and compact [14]. The oxidation resistance temperature of the coating can be increased to 800–850 °C [15,16]. However, it is difficult to form the $TiAl_3$ phase layer by directly adding silicon because Ti and Si tend to react first to generate the Ti-Si intermediate phase [17], thus changing the reaction path between Ti, Al and Si. Therefore, figuring out how to introduce silicon after forming the $TiAl_3$ phase layer preferentially has become a key problem in optimizing the Al-Si coating of titanium alloys.

Therefore, this paper proposes a Self-generated Gradient Hot-dipping Infiltration (SGHDI) method to achieve the Ti-Al-Si gradient coatings after the formation of the $TiAl_3$ phase layer preferentially and the introduction of silicon [18]. The characteristics of the coatings are as follows: the Ti-Al-Si multiphase layer structure based on a Si concentration gradient is formed from the substrate to the outside: The dense $Ti(Al,Si)_3$ alloy layer of solid solution Si atoms, the dispersed bulk of $\tau_2$: $Ti(Al_xSi_{1-x})_2$ phase + *L*-(Al,Si) phase and *L*-(Al,Si) phase. However, a dispersed $\tau_2$ phase results in the non-denseness of the $Al_2O_3$ layer during a high-temperature oxidation process, which reduces the high-temperature oxidation resistance of the coating.

Adding rare earth elements (Ce, La, etc.) is considered to be an effective means to improve the high-temperature oxidation resistance and corrosion resistance of the coating [19]. Ce and La are enriched at grain boundaries, which can effectively reduce the high-temperature oxidation rate [20]. Rare earth elements can also refine the microstructure and structure of coatings [21]. Therefore, adding cerium is proposed to suppress the generation of the $\tau_2$ phase and improve the high-temperature oxidation resistance of the coating. The effect of hot infiltration time on the microstructure and morphology of the Ti-Al-Si gradient coating added with Ce and the high-temperature oxidation resistance of the sample with excellent coating structure were studied.

## 2. Coating Preparation Method and Experiment Method

### 2.1. Coating Preparation Method

The essence of hot-dipping is a liquid/solid diffusion coupling reaction. When the titanium alloy is hot-dip aluminized, a Ti-Al binary intermetallic compound phase layer will be formed on the surface of the titanium alloy. When Si is added, it is difficult to form the $TiAl_3$ phase with excellent high-temperature oxidation resistance because the Ti atoms combine with Si atoms preferentially to form a Ti-Si binary intermediate phase. According to the ternary phase diagram of Ti-Al-Si, the Si atoms can solubilize in the $TiAl_3$ phase and form the secondary solid solution of $Ti(Al,Si)_3$ [22]. Therefore, the $Ti(Al,Si)_3$ phase can be

used as the medium for Si atoms to dissolve into the phase layer of Ti-Al alloy. Therefore, it is necessary to develop a new coating preparation method: the $TiAl_3$ phase layer is formed on the surface of the titanium alloy by hot-dip aluminizing, and then Si atoms are added into aluminum melt and Si atoms are dissolved into the $TiAl_3$ phase layer. Obviously, Si atoms will form a concentration gradient distribution in the $TiAl_3$ phase layer. Then, the key issue is how to introduce Si sources. We use the phenomenon that the aluminum melt can react with quartz glass to generate Si atoms and use the quartz glass tube as a container for pure aluminum liquid, thus cleverly solving the Si sources problem. The new coating preparation method is called the Self-generated Gradient Hot-dipping Infiltration (SGHDI) method.

### 2.2. Experiment Method

A 99.995 wt.% pure Al, Al-20 wt.% Ce master alloy, high-purity quartz glass tube (providing active Si atoms source and acting as a container), and 10-mm-diameter Ti-6Al-4V (TC4) alloy rod were selected for the experiment. The hot infiltration process is as follows:

(1)  Degrease the surface of the TC4 alloy with a metal cleaning agent and dry it;
(2)  Melt the high-purity aluminum in the vertical pit furnace;
(3)  Pour the high-purity aluminum melt in step (2) into the high-purity quartz tube, keep the melt state at a high-temperature of 800 °C, immerse the TC4 alloy into the high-purity aluminum melt, hold for a certain time, quickly extract from the melt, and then quench.

The Al-1 wt.% Ce solution was prepared by adding a moderate amount of Al-20 wt.% Ce master alloy into the high-purity aluminum melt so as to achieve the purpose of adding Ce.

The high-temperature oxidation experiment involves placing coated/uncoated specimens, respectively, in small corundum crucibles, weighing, recording, and then putting them in the box furnace (Xiangtan Samsung Instrument Co., LTD, Xiangtan, China), heating up to 800 °C, taking out a specimen at regular intervals, weighing and recording in order to calculate how much weight is added per unit area at different times, and obtaining the high-temperature oxidation weight curves of coated/uncoated specimens.

The microstructure, phase constitute, and element distribution of the coating were characterized and measured by SEM (ZEISS EVO MA10, Zeiss, Jena, Germany), EDS (OXFORD X-MAXN, Zeiss, Jena, Germany).

## 3. Results and Discussion

### 3.1. Microstructure Characterization of Unadded Cerium Coating

Figure 1 shows the microstructures of the Al-Si coating of TC4 at 800 °C for different hot-dipping times. The dipping coating has obvious delamination, which can be divided into two regions: (1) the surface layer is a liquid phase, *L*-(Al, Si); (2) the alloy phase layer of mutual diffusion between substrate and liquid phase: the dense alloy phase layer, which metallurgically combined with the substrate, and the dispersed bulk alloy phase layer.

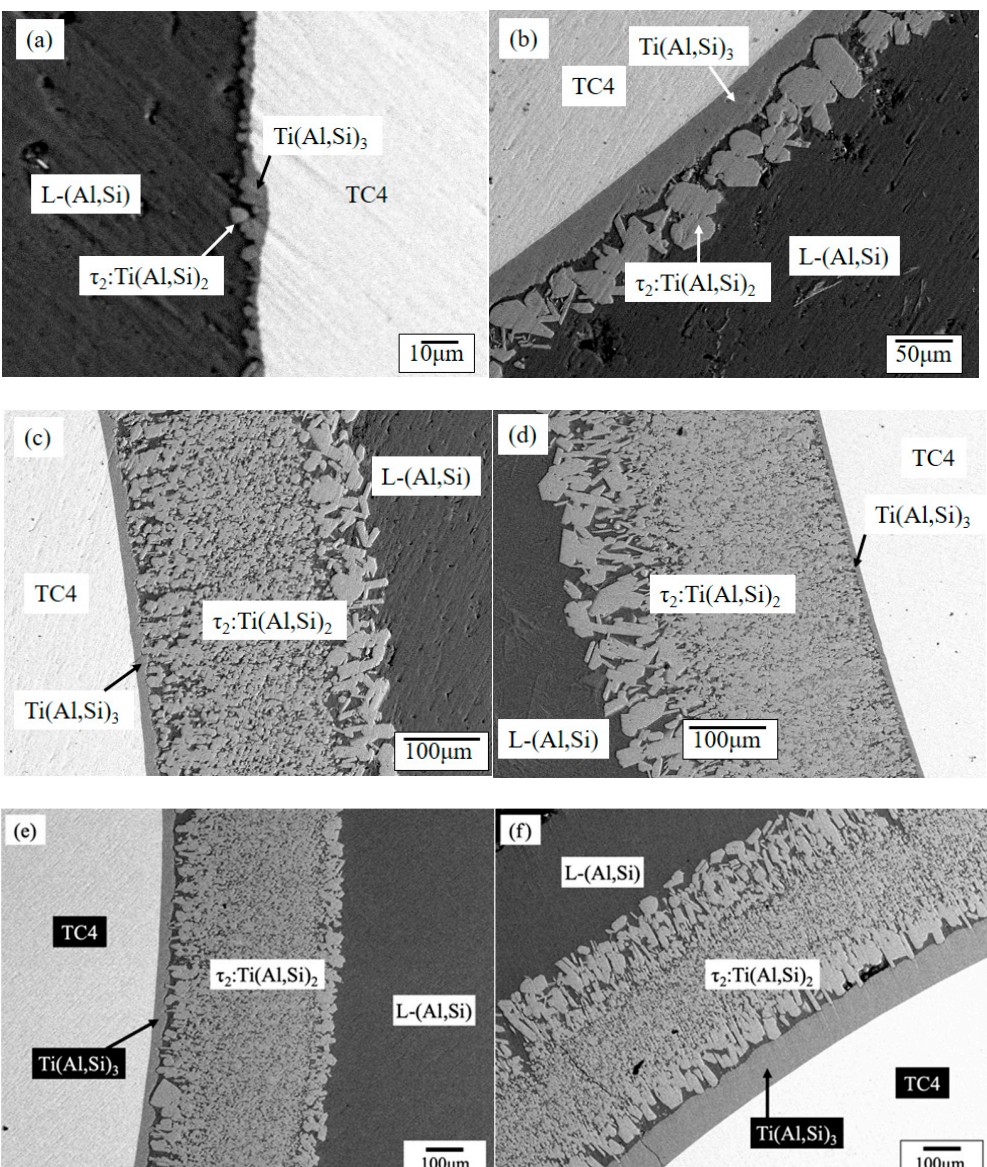

**Figure 1.** The microstructures of Ti-Al-Si gradient coatings on TC4 alloy at 800 °C, (**a**) 10 min, (**b**) 30 min, (**c**) 50 min, (**d**) 80 min, (**e**) 90 min, (**f**) 110 min.

The EDS results show that the elemental atomic ratio is within the range of Ti(Al,Si)$_3$ phase (Al:Si:Ti = 59:14:27) in the dense alloy phase layer, so it is a Ti(Al,Si)$_3$ phase. The EDS results show that the elemental atomic ratio is within the range of the $\tau_2$: Ti(Al,Si)$_2$ phase (Al:Si:Ti = 55:12:33) in the dispersed bulk alloy phase layer, so it is the $\tau_2$: Ti(Al,Si)$_2$ phase. In other words, the Ti(Al,Si)$_3$ phase, $\tau_2 + L$-(Al,Si) phase and $L$-(Al,Si) phase are formed in sequence on the surface of titanium alloy.

Figure 1a shows that although the Ti(Al,Si)$_3$ phase layer is only a few microns in the early stage of the hot-dip aluminizing, it shows that this method realized the preferential reaction between Ti and Al to generate the TiAl$_3$ phase layer. Moreover, once the TiAl$_3$ phase layer is formed, a barrier is constructed to isolate Si atoms from directly contacting the surface of titanium alloy, and the reaction between Ti and Si is effectively prevented. In the subsequent long hot-dipping process (Figure 1a–f), although Si atoms can form an Ti(Al,Si)$_3$ phase by dissolving into an TiAl$_3$ phase, the diffusion rate of Al atoms in the liquid phase through the TiAl$_3$ phase layer to the surface of titanium alloy is much faster than that of Si atoms through the TiAl$_3$ phase layer. Therefore, the formation of a dense and thick Ti(Al,Si)$_3$ phase layer is guaranteed.

The morphology of the $\tau_2$ phase layer is very interesting. The bulk $\tau_2$ phase is uniform in the microstructure of hot-dipping coating (Figure 1a,b). However, with the increase in hot-dipping time, the $\tau_2$ phase layer creates the stratification phenomenon of bulk size scale: inner large block $\tau_2$, middle tiny block $\tau_2$, and outer large block $\tau_2$. The loose bulk $\tau_2$ phase destroys the compactness of the $Al_2O_3$ aluminum layer, and the sharp corner also becomes the source of cracks, so it may adversely affect the overall oxidation-resistance of the coating.

### 3.2. The Microstructure Characterization of Added Cerium Coating

The microstructures of Ce-added hot infiltration coating are shown in Figure 2. With the increase in hot infiltration time, the thickness of the alloy phase layer gradually increased. When the hot infiltration time is less than 50 min, there is a very dense alloy layer, and the liquid phase covers it in the coating. The dense layer is the $Ti(Al,Si)_3$ phase layer confirmed by energy spectrum analysis. When the hot infiltration time reaches 60 min or even higher, the longer the hot infiltration time, the more the bulk phase appears in the liquid phase, which is the $\tau_2$ phase by energy spectrum analysis. The $Ti(Al,Si)_3$ phase layer is gradually thinned due to the appearance of the $\tau_2$ phase. It can be seen that the addition of Ce can suppress the generation of the $\tau_2$ phase in a certain time range.

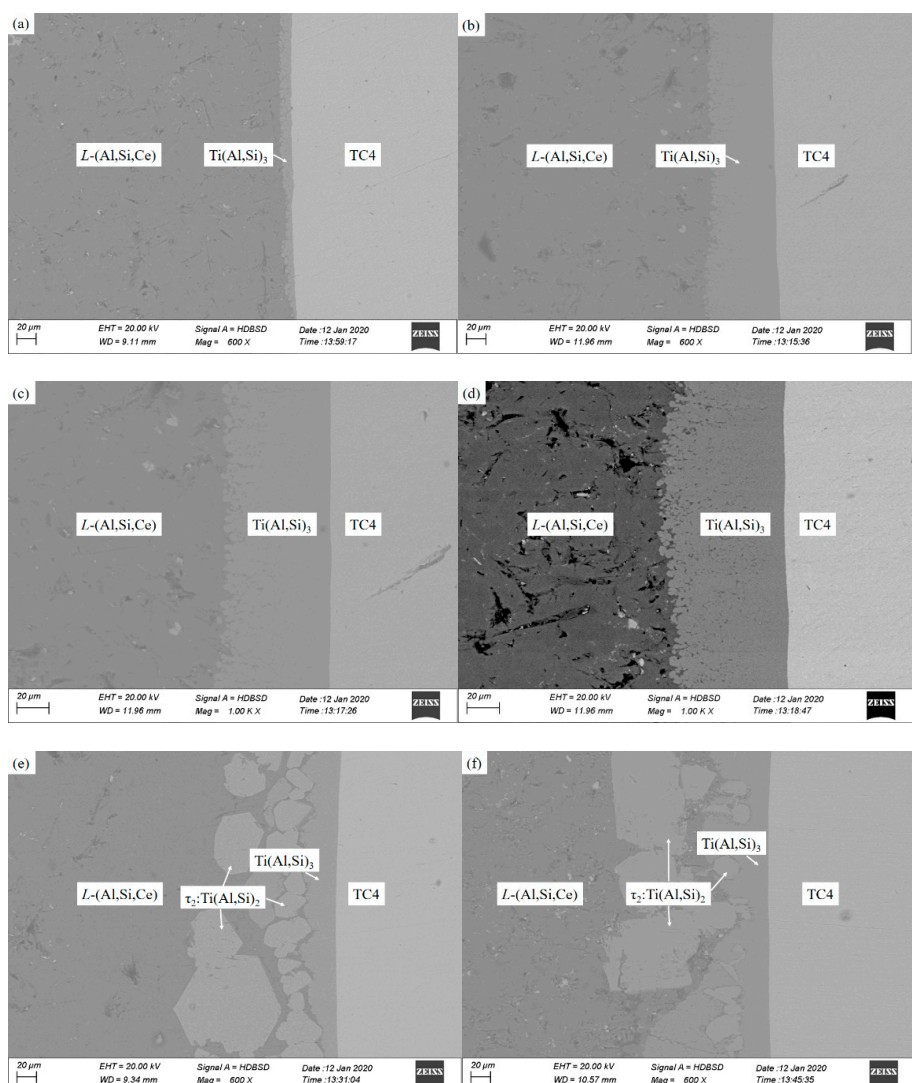

**Figure 2.** The microstructure of the Ce-added coating at different hot infiltration times, (**a**) 20 min; (**b**) 30 min; (**c**) 40 min; (**d**) 50 min; (**e**) 60 min; (**f**) 80 min.

The dense Ti(Al,Si)$_3$ alloy phase layer has excellent resistance to high-temperature oxidation and is the only single phase alloy phase layer in the hot infiltration process. Revealing the growth kinetics of the Ti(Al,Si)$_3$ alloy phase layer is helpful in guiding the structural design of the coating and adjust the high-temperature oxidation resistance of the coating. Considering that the thickness of the Ti(Al,Si)$_3$ alloy phase decreases instead of increasing after the hot infiltration time exceeds 50 min, this paper only analyzes the relationship between the thickness and time of the Ti(Al,Si)$_3$ alloy phase layer within 50 min. There exists an empirical formula [23]:

$$\delta = kt^n \tag{1}$$

where $\delta$ is the thickness ($\mu$m), $t$ is time (min), $k$ is the rate constant, and $n$ is the kinetic exponent. The $n \leq 0.5$ means diffusion-controlled growth, and $n > 0.5$ means reaction-controlled growth.

The growth kinetics curve of the Ti(Al,Si)$_3$ alloy layer was fitted by Formula (1) (as shown in Figure 3):

$$\delta = 0.216t^{1.475} \tag{2}$$

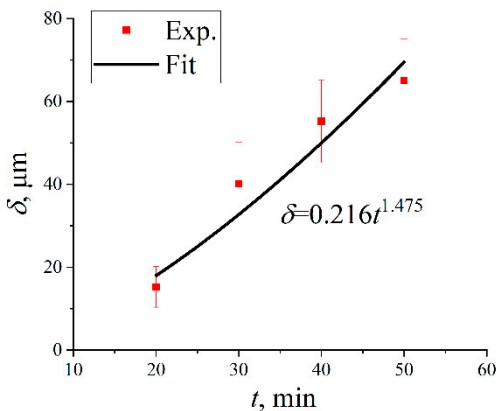

**Figure 3.** The growth kinetics curve of the Ti(Al,Si)$_3$ alloy layer.

The results show that it is reaction-controlled growth.

### 3.3. The Microstructure Characterization of Added Cerium Coating

No Ce compounds are found in the Ti-Al-Si gradient coating, but the effect of adding Ce is obvious, and the growth of the $\tau_2$ phase is inhibited within 50 min of hot infiltration. According to the formation sequence of various phases, the coating growth stage can be divided into four important stages (Figure 4): the formation of the TiAl$_3$ phase, the growth of the TiAl$_3$ phase and Si atom solution, the growth of the Ti(Al,Si)$_3$ phase, and the formation and growth of the $\tau_2$ phase.

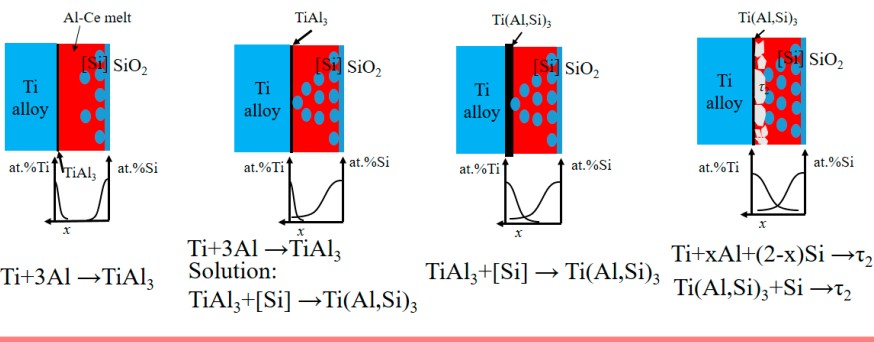

**Figure 4.** Schematic diagram of the growth mechanism of cerium-added coating.

When the TC4 alloy is immersed into Al-Ce melt, titanium atoms in TC4 diffuse to Al melt, and the TiAl$_3$ phase is formed firstly at the interface. The reaction equation is:

$$Ti + 3Al \rightarrow TiAl_3 \tag{3}$$

The Al melt on the quartz tube's wall reacts with SiO$_2$ at the same time, and the generated Si atoms diffuse toward TC4. As the diffusion time increases, solid Si atoms dissolve into TiAl$_3$, and the Ti(Al,Si)$_3$ phase is formed. The solution process is:

$$TiAl_3 + 3[Si] \rightarrow Ti(Al,Si)_3 \tag{4}$$

When Si in TiAl$_3$ is saturated, the Ti, Al and Si atoms at the front of the solid–liquid interface will react to form the Ti(Al,Si)$_2$ phase ($\tau_2$ phase). In addition, the saturated Ti(Al,Si)$_3$ phase reacts with Si atoms to form the $\tau_2$ phase [24]. The $\tau_2$ phase is formed continuously, and the longer the diffusion time, the thicker the $\tau_2$ phase layer and the thinner the Ti(Al,Si)$_3$ phase. The reaction equation to form $\tau_2$ is:

$$Ti + xAl + (2 - x)Si \rightarrow \tau_2 \tag{5}$$

$$Ti(Al,Si)_3 + Si \rightarrow \tau_2 \tag{6}$$

*3.4. High-Temperature Oxidation Resistance*

The results can be obtained by fitting the weight curves of uncoated TC4 substrate alloy samples and the 60 min Ti-Al-Si hot-dipping samples by an isothermal oxidation comparison test at 800 °C in static air,

$$\Delta m = 2.107t^{0.7} \text{ (uncoated)} \tag{7}$$

$$\Delta m = 0.624t^{0.5} \text{ (Ti-Al-Si gradient coating)} \tag{8}$$

where $\Delta m$ is the amount of oxidation weight gain (mg·cm$^{-1/2}$), and $t$ is the oxidation time (h). It can be known from Formulas (7) and (8) that the mass increase rate of the uncoated samples is obviously more rapid than that of the coated samples. The high-temperature oxidation resistance of the Ti-Al-Si coating is better than that of the pure aluminum coating [13], and about the same as the oxidation resistance of the Al-Si coating prepared by Zhou W, et al. [25] by the low oxygen partial pressure self-fusing method. Therefore, this hot-dipping Ti-Al-Si gradient coating has excellent high-temperature oxidation resistance.

Figure 5a shows the isothermal oxidation microstructure of the 60 min hot-dipping sample. The outermost phase layer of the sample after the oxidation is $\tau_2$: the Ti(Al,Si)$_2$ + $\alpha$-Al$_2$O$_3$ phase. The loose $\tau_2$ phase distribution causes the $\alpha$-Al$_2$O$_3$ layer to become incompact. In the isothermal oxidation process, Al and Si atoms in the Ti(Al,Si)$_3$ phase layer diffuse to the substrate and react with Ti atoms to form some new dense phase layers, as shown in Figure 5b. Through energy spectrum analysis, Ti$_3$Al, TiAl, and Ti$_5$Si$_3$ are newly formed. From the distribution of these new phase layers, the diffusion rate of the Al atom is obviously higher than that of the Si atom. The Si atoms are mainly distributed in the original Ti(Al,Si)$_3$ layer, and part of the Si atoms react with Ti atoms to generate Ti$_5$Si$_3$ [26–28].

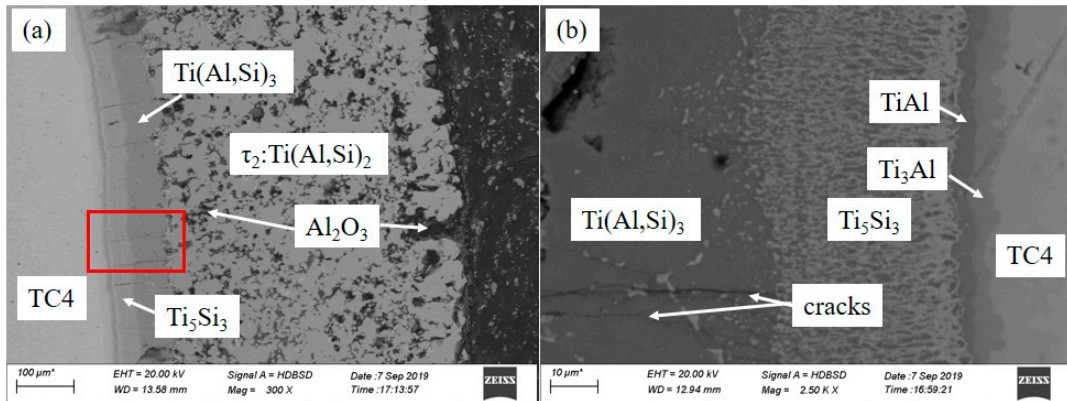

**Figure 5.** (**a**) Microstructure of the pure Ti-Al-Si gradient coating after oxidation for 120 h, (**b**) local magnification of red wireframe in (**a**).

After oxidation for 120 h, microcracks appeared in the Ti(Al,Si)$_3$ phase layer. However, the newly formed dense layer effectively prevented the crack propagation and prevented oxygen from entering the TC4 substrate, which has a good protective effect on the TC4 substrate.

Considering that the hot infiltration time to inhibit the generation of the $\tau_2$ phase is less than 50 min, and the thickness of the Ti(Al,Si)$_3$ phase layer is less than 20 μm when the hot infiltration time is less than 20 min, and the long hot infiltration time is not suitable for industrial production, the Ti-Al-Si coating formed by a hot infiltration time of 30 min is more appropriate. In this paper, the sample with 30 min hot infiltration time was selected for the high-temperature oxidation experiment to analyze its high-temperature oxidation resistance. Figure 6a shows the experiment results of isothermal oxidation for 72 h cerium-added hot-dipping sample for 30 min in static air at 800 °C. The oxidized coating has seven distinct phase layers, and there are many white granular phases in the sixth and seventh phase layers. EDS (shown in Table 1) shows that Ti:Al = 3:1 in the 1# phase layer, so it is Ti$_3$Al. In the 2# phase layer, Ti:Al = 2:1, so it is Ti$_2$Al. In the 3# phase layer, Ti:Al = 1:1, so it is TiAl. In the 4# phase layer, Ti:Al = 1:2, so it is TiAl$_2$. The 5# phase layer is different from 1–4# phase layers in that there exists an oxygen element, and Ti:Al = 1:3, so it is the TiAl$_3$ phase layer. There are only Al, Si, Ti and O elements in the 6# phase layer, which are Ti(Al,Si)$_3$ and Al$_2$O$_3$ phases. There are five elements, Al, Si, O, Ti and Ce in the 7# phase layer, among which 85.34 at.% Al, 10.01 at.% O, 4.35 at.% Si, 0.09 at.% Ti and 0.21 at.% Ce in the gray area. The substrate of the 7# phase layer is $L$-(Al, Si, Ce) + Al$_2$O$_3$. The Al, Si, O, Ti and Ce in the white 8# phase layer are 57.17, 14.08, 3.77, 8.77 and 16.21 at.%, respectively, indicating that it is an Al$_6$Si$_2$Ce$_2$Ti compound. In addition, although cracks can run through the liquid phase layer and Ti(Al,Si)$_3$ layer, they will still be blocked by the newly formed Ti-Al binary layer and cannot penetrate the entire coating.

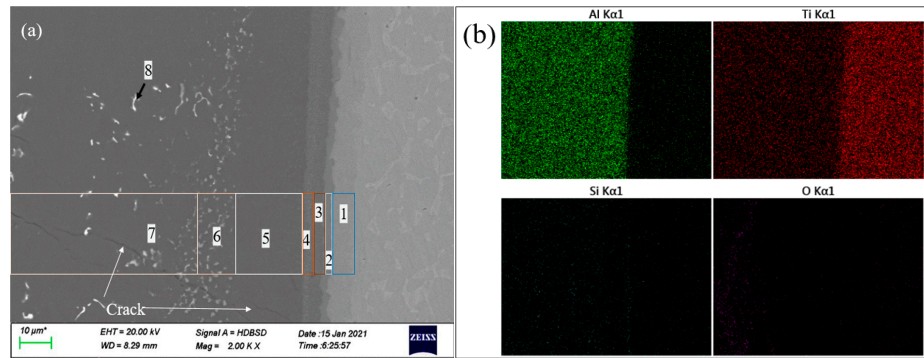

**Figure 6.** Microstructure (**a**) and EDS mapping date (**b**) of adding a cerium Ti-Al-Si gradient coating after 72 h oxidation.

**Table 1.** EDS data and main compounds of each alloy phase layer.

| Phase Layer in Figure 6 | Ti | Al | Si | O | Ce | Compounds |
|---|---|---|---|---|---|---|
| 1# | 76.46 | 23.54 | 0 | 0 | 0 | $Ti_3Al$ |
| 2# | 64.76 | 35.24 | 0 | 0 | 0 | $Ti_2Al$ |
| 3# | 47.40 | 52.60 | 0 | 0 | 0 | TiAl |
| 4# | 34.47 | 65.53 | 0 | 0 | 0 | $TiAl_2$ |
| 5# | 25.43 | 67.07 | 0 | 7.50 | 0 | $TiAl_3$ |
| 6# | 24.48 | 54.90 | 11.95 | 8.67 | 0 | $Ti(Al,Si)_3 + Al_2O_3$ |
| 7# | 0.09 | 85.34 | 4.35 | 10.01 | 0.21 | $L\text{-}(Al, Si, Ce) + Al_2O_3$ |
| 8# | 8.77 | 57.17 | 14.08 | 3.77 | 16.21 | $TiAl_6Si_2Ce_2$ |

Figure 6b shows the distribution of the elements in Figure 6a. The Al element is mainly distributed in the coating, while the Ti element has a small distribution in the coating, indicating that the Ti element has diffused into the coating. There are trace distribution of Si and O elements in the 5#–7# phase layer, indicating that the Ti-Al-Si coating added with Ce inhibits the formation of the loose $\tau_2$ phase layer and greatly improves the high-temperature oxidation resistance.

Although Al, Si and Ti elements all diffuse each other during the high-temperature oxidation process, compared with the Ti-Al-Si coating without Ce added, first, there is no binary Ti-Si phase in the $Ti(Al,Si)_3$ phase layer, and the $Ti(Al,Si)_3$ phase layer remains dense. Second, $Al_6Si_2Ce_2Ti$ compounds are precipitated from the $L$-(Al, Si, Ce) alloy layer during the high-temperature oxidation process, and the concentration of Si, Ce and Ti is much higher than the concentration of its own elements in the surrounding liquid phase, which restrains the generation of the $\tau_2$ phase. Third, the continuous liquid layer facilitates further oxidation into a continuous $Al_2O_3$ phase layer.

*3.5. The Formation Mechanism of High-Temperature Oxidation Microstructure of Adding Ce Coatings*

Figure 7 is a schematic diagram of the high-temperature oxidation mechanism of Ce-added Ti-Al-Si gradient coating with 30 min hot infiltration. Before high-temperature oxidation, the coating consisted of a dense $Ti(Al,Si)_3$ phase layer and a liquid phase. After high-temperature oxidation, oxygen diffuses toward the substrate, forming dense $Al_2O_3$ in the liquid phase, the Al atoms in the $Ti(Al,Si)_3$ diffuse toward the substrate and form a new dense alloy phase layer $TiAl_3$. When the high-temperature oxidation time increases, several new Ti-Al alloy phase layers ($Ti_3Al$, $Ti_2Al$, TiAl, $TiAl_2$) are formed between $TiAl_3$ and the substrate, which can effectively prevent the diffusion of oxygen and crack towards the substrate.

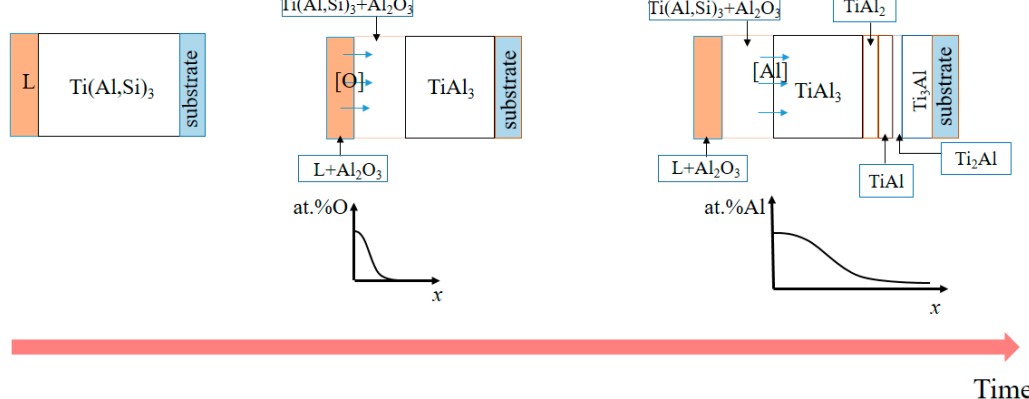

**Figure 7.** Schematic diagram of the high-temperature oxidation mechanism.

## 4. Conclusions

In this paper, a Ti-Al-Si gradient coating was successfully prepared by a self-generated gradient hot-dipping method, and the generation of $\tau_2$ was suppressed by adding Ce. The formation mechanism and high-temperature oxidation resistance of the coating were discussed. The main conclusions are as follows:

(1) The thickness of the Ti(Al,Si)$_3$ phase with a Ce-added Ti-Al-Si gradient coating increases gradually with the increase in hot infiltration time, and the $\tau_2$: Ti(Al,Si)$_2$ phase appears and grows when the hot infiltration time exceeds a certain point.

(2) Within 50 min of hot infiltration, the dense Ti(Al,Si)$_3$ phase layer and $L$-(Al, Si, Ce) phase layer are formed outward from the substrate, respectively. When the hot infiltration time is extended, the dense Ti(Al,Si)$_3$ phase, $L$-(Al, Si, Ce) phase + bulk $\tau_2$: Ti(Al,Si)$_2$ phase and $L$-(Al, Si, Ce) phase are obtained from the substrate in sequence.

(3) Several new Ti-Al system alloy phase layers (Ti$_3$Al, TiAl and TiAl$_3$, etc.) are formed between the substrate and Ti(Al,Si)$_3$ during the high-temperature oxidation process, which can further prevent the diffusion of oxygen and cracks to the substrate.

(4) Adding Ce can form Ce-rich quaternary phase (TiAl$_6$Si$_2$Ce$_2$) in $L$-(Al, Si, Ce) alloy layer and suppress the formation of the Al-Si-Ti ternary phase and $\tau_2$: Ti(Al,Si)$_2$ phase. The concentration of Ti, Si and Ce in the TiAl$_6$Si$_2$Ce$_2$ phase is much higher than that of these elements in the surrounding $L$-(Al, Si, Ce) alloy, indicating that the element segregation effect occurs after the addition of Ce.

The hot infiltration method is an economical and efficient method for preparing Al-based coatings for high-temperature-resistant titanium alloy. As the α-β phase transition temperature of titanium alloy is about 900 °C, the hot infiltration temperature cannot exceed 900 °C. The key to high-temperature oxidation resistance is to form a dense and continuous multiphase gradient coating, which is to isolate air and avoid cracks caused by thermal stress between layers. A Ti-Al-Si gradient coating added with Ce can inhibit the formation of a loose $\tau_2$ layer and maintain a dense and continuous multiphase layer structure in the high-temperature oxidation process for a long time. Oxygen atoms cannot diffuse into the substrate, and the Ti-Al-Si gradient coating showed excellent high-temperature oxidation resistance. This work provides a new idea for the preparation and structure design of hot dipping aluminum alloy coatings.

**Author Contributions:** Data curation, W.H. and Y.T.; Investigation, X.H.; Methodology, Z.L.; Project administration, F.L.; Writing—original draft, Z.W. All authors have read and agreed to the published version of the manuscript.

**Funding:** This research was funded by the Hunan Provincial Natural Science Foundation of China (No. 2021JJ30672), National Natural Science Foundation of China (No. 52171017), Research initiation project of Xiangtan University and College Students' innovation and entrepreneurship training program of Xiangtan University.

**Institutional Review Board Statement:** Not applicable.

**Informed Consent Statement:** Not applicable.

**Data Availability Statement:** Not applicable.

**Acknowledgments:** The authors gratefully acknowledge the support provided by Fucheng Yin and Key Laboratory of Materials Design and Preparation Technology of Hunan Province, Xiangtan University, Xiangtan, Hunan.

**Conflicts of Interest:** The authors declare no conflict of interest.

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
