# Peer review of "Preparation of Ti-Al-Si Gradient Coating Based on Silicon Concentration Gradient and Added-Ce"

_coatings, doi:10.3390/coatings12050683_

Round 1
Reviewer 1 Report
This is a nice effort and presents some unique results. However, the authors appear to be struggling with the role of cerium which is the major contributor to densification. At least in two places, they have used language that indicates liquid phase clustering? I was not clear about this.
- Second, Al6Si2Ce2Ti compounds appear in the liquid phase line 246
- Quaternary phase in liquid phase in Conclusions.
Perhaps the authors can clarify this some more. It is extremely unclear what they are intending to say here.
Author Response
Comments and Suggestions for Authors
This is a nice effort and presents some unique results. However, the authors appear to be struggling with the role of cerium which is the major contributor to densification. At least in two places, they have used language that indicates liquid phase clustering? I was not clear about this.
- Second, Al6Si2Ce2Ti compounds appear in the liquid phase line 246
- Quaternary phase in liquid phase in Conclusions.
Perhaps the authors can clarify this some more. It is extremely unclear what they are intending to say here.
Response: Thanks for the reviewer’s kind suggestion. We changed the "liquid phase" to "L-(Al, Si, Ce) alloy layer", so that it is clear that the Al6Si2Ce2Ti compounds is precipitated from L-(Al, Si, Ce) alloy layer during the high-temperature oxidation process.
The red font in the revised manuscript is modified according to the opinions of reviewers.
Reviewer 2 Report
This manuscript reports on preparation of Ti-Al-Si gradient coating based on silicon concentration gradient and Ce doping modification. Paper could not be accepted in the presented way. Results should be better structured and summarized. Experimental part should be expanded and fully described. All old results could be used if they are used for comparison. At the moment previous results and new results are not connected. Reader should understand why these or another results are presented. XRD, EDX mapping and oxidation curves of Ti-Al-Si gradient coatings doped with Ce should be added and described. Influence of cerium is not clearly written. I recommend a major revision.
- From introduction, it is not understandable why cerium was added to Ti-Al-Si coating.
- Page 1 Line 29 should be “aero” instead of “areo”.
- I am kindly asking to distinguish usage of previously published paper in this work https://doi.org/10.1016/j.jallcom.2020.154670, since Figure 2, 3, 6, 7, 13 were taken from this paper with practically the same description. Authors could use previously published results, but this should be correspondingly cited and with rights for reprinting from Elsevier.
- In 4.3 Ce is not mentioned in the description.
- Why is there no growth kinetics curves of Ti-Al-Si doped Ce? In Fig. 8 EDX mapping is required. It is also not understandable representation of only cerium-doped sample after hot dipping for 30 min. EDS data of alloy phase layers should be added as well.
Author Response
Comments and Suggestions for Authors
This manuscript reports on preparation of Ti-Al-Si gradient coating based on silicon concentration gradient and Ce doping modification. Paper could not be accepted in the presented way. Results should be better structured and summarized. Experimental part should be expanded and fully described. All old results could be used if they are used for comparison. At the moment previous results and new results are not connected. Reader should understand why these or another results are presented. XRD, EDX mapping and oxidation curves of Ti-Al-Si gradient coatings doped with Ce should be added and described. Influence of cerium is not clearly written. I recommend a major revision.
- From introduction, it is not understandable why cerium was added to Ti-Al-Si coating.
Response: Thanks for the reviewer’s kind suggestion. We have added the reason why Ce is doped in the introduction. "Adding rare earth elements (Ce, La, etc.) is considered to be an effective means to improve the high-temperature oxidation resistance and corrosion resistance of the coating [19]. Ce and La are enriched at grain boundaries, which can effectively reduce the high-temperature oxidation rate [20]. Rare earths can also refine the microstructure and structure of coatings [21]. Therefore, cerium doping is proposed to suppress the generation of τ2 phase and improve the high-temperature oxidation resistance of the coating. The effect of hot infiltration time on the microstructure and morphology of Ti-Al-Si gradient coating doped with Ce, and the high-temperature oxidation resistance of the sample with excellent coating structure were studied."
- Page 1 Line 29 should be “aero” instead of “areo”.
Response: Thanks for the reviewer’s kind suggestion. We corrected the spelling.
- I am kindly asking to distinguish usage of previously published paper in this work https://doi.org/10.1016/j.jallcom.2020.154670, since Figure 2, 3, 6, 7, 13 were taken from this paper with practically the same description. Authors could use previously published results, but this should be correspondingly cited and with rights for reprinting from Elsevier.
Response: Thanks for the reviewer’s kind suggestion. We selected some unpublished images and rewrote the introduction to the previous work to avoid a simple repetition of the previously published paper.
- In 4.3 Ce is not mentioned in the description.
Response: Thanks for the reviewer’s kind suggestion. We added a new discussion in section 3.3 of the modified version. "No Ce compounds are found in the Ti-Al-Si gradient coating, but the effect of doping Ce is obvious, and the growth of τ2 phase is inhibited within 50 min hot dipping time. According to the formation sequence of various phases, the coating growth stage can be divided into four important stages: formation of TiAl3 phase, growth of TiAl3 phase and Si atom solution, growth of Ti(Al,Si)3 phase, and formation and growth of τ2 phase."
- Why is there no growth kinetics curves of Ti-Al-Si doped Ce? In Fig. 8 EDX mapping is required. It is also not understandable representation of only cerium-doped sample after hot dipping for 30 min. EDS data of alloy phase layers should be added as well.
Response: Thanks for the reviewer’s kind suggestion. The growth kinetics curves of Ti(Al,Si)3 alloy phase layer doped Ce is analyzed in the revised paper.
We add the Fig. 8 EDS mapping date in Fig. 6(b) of the revised paper. The distribution of each element is also discussed.
We add a paragraph to explain why we only study the high temperature oxidation performance of samples with 30 min hot dipping time. "Considering that the hot dip time to inhibit the generation of T phase is less than 50 minutes, and the thickness of TiAl3 phase layer is less than 20 microns when the hot dip time is less than 20 minutes, and the long hot dip time is not suitable for industrial production, the Ti-Al-Si coating formed by hot dip time of 30 minutes is more appropriate. In this paper, the sample with 30 min hot dip time was selected for high temperature oxidation experiment to analyze its high temperature oxidation resistance."
We add table 1 to show the EDS data of alloy phase layers.
The red font in the revised manuscript is modified according to the opinions of reviewers.
Reviewer 3 Report
The article under review is devoted to a hot topic, namely, the increase in corrosion properties at high temperatures of the TC4 alloy. The article is well-written, well-structured and has high-quality graphic content. The work has practical significance and a scientific component. At the same time, I can make some recommendations to the authors of the article and I think that these comments will improve the article even more.
- Lines 74-83 attempt to state the purpose of the study. However, in my opinion, now the purpose of the work is not formulated very clearly. I recommend the authors to more clearly formulate the purpose of the study and its scientific novelty.
- Figure 6 (c) is too small and of poor quality. I recommend making it a separate figure, increasing and increasing the resolution.
- Figure 7 (b) must also be enlarged. It's hard to read.
In the conclusions of the study, the authors should more clearly and specifically state the practical recommendations for coating applications for industry. To specify the processing modes for obtaining a high-quality coating on the TC4 alloy.
Author Response
Comments and Suggestions for Authors
The article under review is devoted to a hot topic, namely, the increase in corrosion properties at high temperatures of the TC4 alloy. The article is well-written, well-structured and has high-quality graphic content. The work has practical significance and a scientific component. At the same time, I can make some recommendations to the authors of the article and I think that these comments will improve the article even more.
- Lines 74-83 attempt to state the purpose of the study. However, in my opinion, now the purpose of the work is not formulated very clearly. I recommend the authors to more clearly formulate the purpose of the study and its scientific novelty.
Response: Thanks for the reviewer’s kind suggestion. We have added the reason why Ce is doped in the introduction. "Adding rare earth elements (Ce, La, etc.) is considered to be an effective means to improve the high-temperature oxidation resistance and corrosion resistance of the coating [19]. Ce and La are enriched at grain boundaries, which can effectively reduce the high-temperature oxidation rate [20]. Rare earths can also refine the microstructure and structure of coatings [21]. Therefore, cerium doping is proposed to suppress the generation of τ2 phase and improve the high-temperature oxidation resistance of the coating. The effect of hot infiltration time on the microstructure and morphology of Ti-Al-Si gradient coating doped with Ce, and the high-temperature oxidation resistance of the sample with excellent coating structure were studied."
- Figure 6 (c) is too small and of poor quality. I recommend making it a separate figure, increasing and increasing the resolution.
Response: Thanks for the reviewer’s kind suggestion .Figure 6 is one of the figures of the article we published before. The chief editor suggested us to make some modifications, so we replaced the relevant pictures and made a new writing.
- Figure 7 (b) must also be enlarged. It's hard to read.
Response: Thanks for the reviewer’s kind suggestion. Figure 7 is one of the figures of the article we published before. The chief editor suggested us to make some modifications, so we replaced the relevant figures and made a new discussion.
In the conclusions of the study, the authors should more clearly and specifically state the practical recommendations for coating applications for industry. To specify the processing modes for obtaining a high-quality coating on the TC4 alloy.
Response: Thanks for the reviewer’s kind suggestion. In the conclusion part, we add the description of the key elements of titanium alloy coating and the positive significance of coating preparation method and coating structure to industrial production.
"It is an economical and efficient method to prepare Al-based coatings for high temperature resistance of titanium alloy by hot infiltration method. As the α-β phase transition temperature of titanium alloy is about 900 oC, the hot infiltration temperature cannot exceed 900 oC. The key of high temperature oxidation resistance is to form a dense and continuous multiphase gradient coating, which is to isolate air and avoid cracks caused by thermal stress between layers. Ti-Al-Si gradient coating doped with Ce can inhibit the formation of loose τ2 layer and maintain a dense and continuous multiphase layer structure in the high-temperature oxidation process for a long time. Oxygen atoms cannot diffuse into the substrate, and the Ti-Al-Si gradient coating showing excellent high-temperature oxidation resistance. This work provides a new idea for the preparation and structure design of hot dipping aluminum alloy coatings."
The red font in the revised manuscript is modified according to the opinions of reviewers.